# Is the digital economy empowering high-quality tourism development? A theoretical and empirical research from China

**Min Yu**[1], **Binbin Ma**[1]*, **Dan Liu**[1], **Aixia Zhang**[2]

**1** School of Tourism, Northwest Normal University, Lanzhou, Gansu, China, **2** School of Tourism and Hotel Management, Dongbei University of Finance and Economies, Dalian, Liaoning, China

* mabinbinly@163.com

**Data Availability Statement:** All relevant data are within the manuscript and its Supporting Information files.

**Funding:** Gansu Provincial Education Science and Technology Innovation Project "Research on the

## Abstract

How digital economy (DE) empowers high-quality development of tourism (HQDT) has become a common concern among scholars. Given this, this study clarifies the theoretical connotation of DE enabling HQDT,and finds that: Micro, DE promotes efficiency improvements in tourism enterprises, with its economies of scale and Matthew effect reducing average costs, its economies of scope meeting diversified demand, and its long-tail effect improving supply-demand matching mechanism. Meso, DE can transform and upgrade tourism industry structure through industrial digitization and digital industrialization, and also form a new tourist industry form and value chain through cross-border integration. Macro, DE can stimulate innovation and flexibility of market players, increase new factor inputs in tourism, improve factor allocation efficiency, and advance macro regulation of the tourism market. Accordingly, the study conducts an empirical test based on panel data for 31 provinces in mainland China during 2011–2020. Results show that: ① DE positively influences HQDT, and the sub-dimensions all positively influence HQDT. ② DE has a heterogeneous impact on HQDT and shows spatial spillover effects. Finally, the study concludes with effective paths for DE promoting HQDT: "Promote digital infrastructure construction, accelerate tourism digital transformation, strengthen integration and innovation development, and overcome the challenges of tourism enterprises".

## 1. Introduction

With the deepening of the new round of industrial and technological revolution, the DE, with digital knowledge and information as resources, digital platforms as carriers, and ICT efficient utilization as benefits, is rapidly emerging [1]. Statistics from the China Academy of Information and Communications Technology show that in 2005, the scale of China's DE was only 2.6 trillion yuan, but by 2022, the scale of China's DE has reached 50.2 trillion yuan, accounting for 41.5% of GDP, with the total volume firmly ranking second in the world. The DE has had a profound impact on the development model and operational structure of all areas of the economy and society. The Chinese government document "Continuously Strengthening and

Innovation and Path of Intelligent Tourism Service Mode in Gansu Province under the Background of Digital Economy" (Grant No. 2022B-325); Gansu Provincial Youth Science and Technology Fund Project "Liupan Mountain Area of Gansu Province: Identification of Rural Regional Types, Spatial Reconstruction and Construction of Rural Revitalization Mode" (Grant No. 21JR7RA156). The sponsor or funder played no role in study design, data collection and analysis, publication decisions, or manuscript writing.

**Competing interests:** NO authors have competing interests Enter: The authors have declared that no competing interests exist.

Expanding China's Digital Economy" points out that the DE is not only a new economic growth point, but also an important engine for building a modernized economic system. At the same time, the Chinese government also clearly pointed out that "high-quality development is the primary task of building a modern socialist country in all aspects". Since 1978, after more than 40 years of development, most of China's modern industries have moved from the high-speed growth stage to the high-quality development stage. Tourism, as an important part of China's modern industries, has also stepped into the stage of high-quality development [2]. High quality development of tourism (HQDT) is a key requirement to make tourism a happy industry, a realistic choice for many years since the construction of China's tourism industry [2], and an important guarantee to meet the growing needs of people for a better life [3]. It has been proven that the DE, as the core driving force of China's economic development, is seen as a major historical opportunity to drive high-quality development [4, 5]. The Chinese government document "The 14th Five-Year Plan for Tourism Development" also clearly states that the DE should be used to promote HQDT [6].

How to effectively unleash the DE to boost the HQDT has become a key action issue for all sectors of society in recent years. Existing studies have found that the DE can boost the high-quality development of manufacturing industry not only through digital industrialization and digital coupling [7], but also through digital transformation [8]. Meanwhile, Chen and Hong argue that the DE can empower high-quality development of agriculture by expanding the scale of agricultural development, improving development efficiency, and increasing development benefits [9], while Lu and Du argue that its empowering role is achieved by means of advanced and rationalized industrial structure [10]. In the service industry, the DE can optimize resource allocation, innovate institutional mechanisms [11], and promote cross-border integration [12] to advance its high-quality development. However, as a comprehensive service industry, does the DE also have an enabling effect on the HQDT? If this effect is confirmed, what are the mechanisms behind it? What are the differences in the spatial effects of the DE on the HQDT? The objective of this paper is to address the questions raised above. There is only a very small body of literature that explores the impact of the DE on HQDT. To answer these questions, we need to sort out the theoretical mechanisms of the DE's effect on HQDT and conduct an empirical study based on the Chinese reality context, which creates an opportunity for this study to make a marginal contribution.

To achieve this goal, this study attempts to answer the theoretical and practical questions about the impact of the DE on the HQDT from both theoretical analysis and empirical testing. First, at the theoretical level, this paper analyzes the impact mechanisms of the DE on the HQDT based on a "micro-meso-macro" framework. Second, at the empirical level, the study selects panel data of 31 Chinese provinces (excluding Hong Kong, Macao, and Taiwan) from 2011 to 2020, and validates the impact of DE on HQDT through benchmark regressions in terms of core variables and their sub-dimensions. The spatial effect of the DE on the HQDT is confirmed through a spatial econometric model; and its endogeneity and robustness are fully tested. Finally, this paper divides the research sample into different provincial sizes and different DE levels to fully test the effect of heterogeneity.

The rest of this study is organized as follows. Section 2 reviews the relevant literature and presents the marginal contributions of this study. Section 3 provides a theoretical exposition on the basis of which the research hypotheses are formulated. Section 4 describes the relevant variables and the research methodology. Section 5 presents empirical results. Section 6 outlines the discussion. Section 7 summarizes the conclusions and management implications, and highlights the limitations of this study and future research directions.

## 2. Literature review

Currently, relevant studies on the HQDT mainly revolve around the relationship between technological innovation [13], industrial integration [14], smart tourism [15–17], tourism resource conversion efficiency [18], tourism competitiveness [19], and the HQDT. Combing the literature reveals that there are two main categories of literature related to the DE for HQDT: ① At the theoretical level: Microscopically, digital intelligence technology can establish and maintain a good relationship between consumers and tourism enterprises by providing high quality services to further improve consumer satisfaction [20]. If tourism enterprises are considered as the supply side, the AI in the growth phase is influencing their employment, costs, and management practices [21]. Dimitra et al. argue that big data can improve the efficiency, productivity, and profitability of tourism enterprises and can provide differentiated, rich, and convenient experiences to consumers [22]. Macroscopically, a part of scholars believe that digital technology can empower the HQDT by enhancing the efficiency of tourism industry, promoting the upgrading of tourism industry structure [1], and promoting tourism business model innovation [23]. Another part of scholars believe that the DE can promote HQDT by changing the market mechanism of tourism development [24] and activating the potential for integrated development of culture and tourism [25]. However, most studies lack a systematic and continuous discussion of the relationship between the DE and the HQDT. They are biased to micro or macro unilateral explanation of the DE's contribution to HQDT, ignoring the fact that the DE for HQDT is a dynamic evolutionary process from micro to meso to macro [1]. ② At the empirical level: Scholars have favored different perspectives and used statistical data to develop their research, e.g., Yang verified that the Internet not only dynamically optimizes the tourism industry but also increases the productivity of tourism enterprises from an industrial economics perspective [26]. Wang et al. constructed a spatial econometric model from the perspective of regional association and argued that information technology can reshape the interaction and association of tourism between regions [27]. However, few studies have considered the spatial effect of DE on the HQDT. As for the evaluation system of HQDT level, Yan and Hu believed that the HQDT level should be measured in seven dimensions: Industrial vitality, innovation, openness, green, coordination, effectiveness, and sharing [28]. Wang et al. argued that the comprehensive measurement of HQDT needs to be cut from 3 aspects: Industry operation, transformation, and sharing [29]. Liu and Tang measured the level of regional HQDT from 6 dimensions: Tourism product service, economic vitality, coordinated development, green development, innovative development, and shared development [30]. It can be seen that the studies about the evaluation of HQDT have formed different systems, and there is no unified index system. The studies about the relationship between DE and HQDT show sporadic and fragmented characteristics, and the research methods at home and abroad are mainly qualitative studies, lacking the combination of qualitative and quantitative studies.

The possible marginal contributions of this study lie in the following aspects: ① Innovation in logical interpretation. Considering that the DE for HQDT is a dynamic evolutionary process, this study constructs a clear explanatory framework from micro to meso to macro levels to explore the relationship between the two in a systematic and continuous manner. ② Improvement in research content. This study integrates the DE and HQDT into the same analysis system and adds the former to explore the spatial effect of the latter, which can further increase the accuracy of the empirical analysis. ③ Improvement in indicator measurement. The Chinese government emphasizes that "we cannot talk about high-quality development without talking about the new development concept. The new development concept must be implemented completely, accurately, and comprehensively." It is the guidance that governs the

whole situation, the root and the long term, and is strategic, programmatic and leading. Therefore, this study constructs an indicator system for HQDT based on the five development concepts, and strives to form an indicator system with rigor, universality, and uniformity. Improvement in research method. This study is not a unilateral qualitative or quantitative study, but a combination of qualitative and quantitative research. The study aims to provide theoretical reference and empirical evidence for deepening the DE to promote the HQDT.

## 3. Theoretical explanation and research hypothesis

### 3.1 Micro level

**3.1.1 Digital infrastructure (DI) helps achieve economies of scale and scope.** Whether traditional tourism enterprises that are undergoing digital transformation or digital emerging tourism enterprises, they all pursue large-scale and diversified operations, which contributes to the supply side of tourism moving closer to high quality [31]. On the one hand, economies of scale indicate that the average cost per unit of an economic agent's product gradually declines as the scale of its production continues to expand [31]. The DE is characterized by value-addedness, diminishing marginal costs, etc., and its Internet domain has been dominated by Metcalfe's law—the value of a network is equal to the square of its number of nodes [32]. Under the effect of network externalities, when the scale of enterprise users reaches a critical capacity, it induces positive feedback, triggers the Matthew effect of the stronger being stronger, and helps economic agents achieve economies of scale [33]. Under the influence of the DE background, traditional tourism enterprises pay more attention to digital transformation, such as continuously increasing the proportion of R&D, timely updating DI systems, and regularly upgrading core equipment.

Digital transformation is not only seen as an important paradigm for the green and low-carbon transformation of enterprises [34], but also as a necessary path for high-quality corporate development [35]. DI is not only an external driver of digital transformation for traditional enterprises [36], but its lack is also closely related to the high mortality rate of entrepreneurship in emerging enterprises [37]. At the same time, based on the substantial improvement of the core technology of DI, enterprises can achieve a larger scale of optimal production and further realize the reduction of average cost. Both supply and demand sides benefit from DI tourism services, for example, consumers can consume tourism products without time and space constraints, saving a large amount of time and space costs. Enterprises can use DI such as the Internet to greatly reduce the human factor, further saving transaction costs, under the positive feedback mechanism to achieve economies of scale on both sides of the supply and demand. Under the two-way interaction between supply and demand, the consumer demand of China's tourism industry is increasing. There are still great opportunities for economic agents to achieve economies of scale, especially for listed companies in scenic spots.

On the other hand, economies of scope are those that result from the scope rather than the scale of a firm's production, namely, the situation that exists when the cost of producing multiple products simultaneously is less than the sum of the costs required to produce each product separately [33]. DI such as mobile technology, the Internet of Things, and automation have enabled the realization of economies of scale while allowing economic individuals to focus on multiple businesses or products. By effectively using the accumulated users of their core business to create diverse products or businesses, tourism companies can not only share their costs but also enrich their profit sources [33]. In the tourism industry, hotel and travel agency businesses have the strongest economies of scope, followed by scenic and non-travel agency businesses [38]. In the era of DE, the demand of tourism consumers is not just singular, standardized, cost-effective and localized, but unitized, personalized, experiential and

virtualized, which also contributes to the long-tail effect on the demand side. The application of DI, while satisfying the diverse needs of consumers, has also led tourism businesses to step into different vibrant areas [20]. Traditional tourism businesses are difficult to achieve sharing because of their minimal association with non-tourism businesses, leading to diseconomies of scope. However, the emergence of DI such as cloud computing, 5G, and big data has enhanced the connection, integration, and interaction of resources and information, enabling the rapid development of the sharing economy and helping to achieve economies of scope.

**3.1.2 DI helps to improve the matching mechanism.** Information technology can use its connectivity function to symbolize supply and demand information, i.e., transform it into data, and then realize effective integration and precise matching through DI, and the allocation efficiency of resources is greatly enhanced. DI construction, as the cornerstone of DE development, not only enables transactions that were previously difficult to obtain matching before the DE era to be carried out smoothly, but also allows transaction prices to gradually approach the Pareto optimal state [31]. Theoretically, this can solve the information asymmetry problem between the supply and demand sides to a certain extent. At the same time, the supply side can respond to the diversified demands on the consumer side in a timely manner, creating an interactive effect between supply and demand and promoting high-quality development. For the tourism industry, tourism destinations can use DI to better understand customers, more accurately segment customers, more innovatively relate to customers, provide more timely feedback to customers, continuously optimize management and services, and realize effective development and green development of tourism destination resources. On a peer-to-peer basis, digital tourism can increase the sharing economy between participants [39]. Gozzi shows that destinations with better quality DI are also more resilient to the impact of mobility cuts during a new coronary pneumonia outbreak [40]. Moreover, with the long-tail effect of acquired data, tourism enterprises can make tourists' experience before, during, and after the tour more colorful and make the matching effect exceed the tourists' expected vision.

Accordingly, the micro-mechanism of DE facilitating HQDT is shown schematically in Fig 1. This paper proposes the following hypothesis:

**Hypothesis 1.** The DE can promote the HQDT through DI construction.

## 3.2 Meso level

**3.2.1 Achieve industrial structure upgrade.** The DE is a catalyst for the upgrading of China's industrial structure, which in turn is an internal driving force for the HQDT in China. The DE mainly contains two parts: Digital industrialization and industrial digitization [41]. On the one hand, industrial digitization refers to the process of industry empowering business upgrading, increasing production quantity, and improving efficiency through digital technology. Industrial digitization mainly uses technology and efficiency improvements to promote industrial structure upgrading. The tourism industry attracts a large number of production factors such as talents and capital with the help of digital technology as a way to increase the share of knowledge and technology-intensive fields, promote high intensification and high technology in the leading industries, increase the synergy between tourism resources and tourism services and other elements, broaden the boundaries of immersive tourism experiences, accelerate the transformation of traditional industries under the tourism industry, and finally achieve the upgrading of industrial structure. While promoting the digital transformation of the tourism industry, the DE can not only realize the enhancement of the allocation efficiency of each element, but also drive the innovation of new industries such as research tourism, folklore tourism, and rural tourism, while also prompting the protection of the environment and

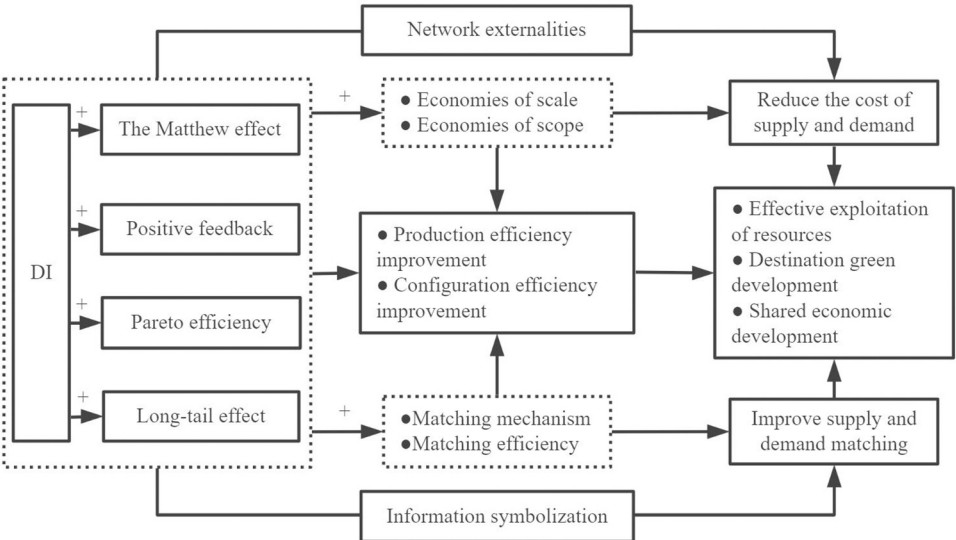

**Fig 1. A schematic diagram of the micro-mechanism of the DE in promoting the HQDT.**

green development. Sustainability of resources can also be achieved through technological innovation under digital transformation [42].

On the other hand, digital industrialization refers to the public industry that takes information as the processing object, digital technology as the processing means, consciousness products as the results, and intervention in all fields of society as the market, which has no obvious profit for itself but can enhance the profit of other industries. The most representative one is the information and communication industry. It uses its strong drive and sense to produce correlation effect, spillover effect, technology diffusion effect, and innovation effect with the tourism industry [43], which establishes its key position in the tourism industry structure and enriches the profit source of the tourism industry, and realizes the tourism industry structure to a higher level. In the context of DE, IT investment has become an important path for digital transformation [44]. Through smart technologies, the tourism industry can overcome the traditional limitations of its transformation [45]. The traditional business model of tourism industry, driven by digital industrialization, realizes that it should not only be centered on creating corporate value, but also on creating customer value.

**3.2.2 Industrial integration and innovation development.** Tourism industry integration refers to the interconnected and interpenetrating relationship that occurs within the tourism industry or between the tourism industry and other industries, ultimately forming a new industrial form. The essence of tourism industry integration lies in innovation, and tourism innovation must be based on certain technical means, especially the development and innovation of digital technology has become a direct driving force of tourism industry integration. Digital technology promotes emerging trends of industry integration such as cross-border integration and collaborative innovation in the tourism industry through the use of its high permeability and versatility. The innovation dilemma can be overcome by the strong innovation quality and strong absorption and transformation capabilities that accompany digital transformation [46]. The cornerstone of the integration of cultural tourism as a typical tourism industry is the integration of technology on both sides. And digital technology can form an industrial linkage effect by helping the tourism industry to integrate with other industries. The combination of resource development technology and GIS technology of the tourism industry

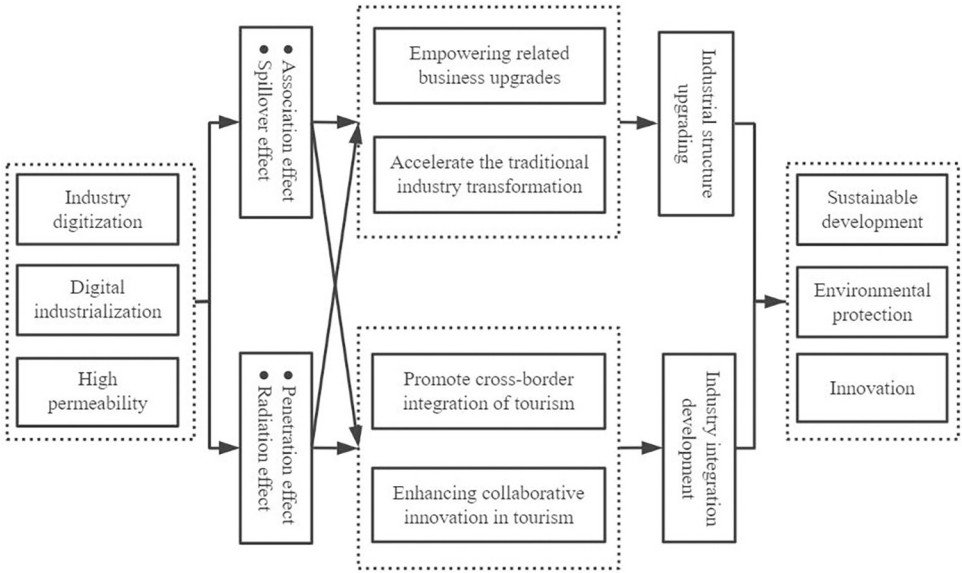

**Fig 2. A schematic diagram of the middle-view mechanism of the DE in promoting the HQDT.**

and item development technology of the cultural industry forms the technical basis for the in-depth integration of both parties. Further, through industrial integration, the cultural industry can use effects such as penetration and radiation to help improve the appreciation of tourism resources, enrich the spiritual and cultural connotations of tourism, and gift the uniqueness of different tourism products, so that tourism is no longer uniform and visitors can feel a better tourism experience. At the same time, DI greatly stimulates the potential inherent in the development of integrated tourism, such as cultural tourism and rural tourism, constantly optimizes and innovates the future shape of tourism, and ensures the sustainable development of tourism in the future [47].

In summary, the middle-view mechanism of DE facilitating HQDT is shown schematically in Fig 2. The following underlying hypotheses are proposed in this paper:

**Hypothesis 2.** Digital industrialization contributes to the realization of HQDT.

**Hypothesis 3.** Digitalization of industry contributes to the realization of HQDT.

**Hypothesis 4.** DE can promote the HQDT through digital innovation.

### 3.3 Macro level

**3.3.1 Enhancing the input of new factors and the efficiency of factor allocation.** Under the gradual and deep integration of digital technology and economic activities, data is no longer merely an auxiliary resource, but gradually becomes independent and evolves into an important factor of production for high-quality development. With the increasing proportion of data-based factors of production, the allocation of factor inputs becomes more rational. With the deeper integration with physical capital and labor, the efficiency of traditional factors of production is gradually improved. With the high synergy and high penetration of the DE, the allocation among factors gradually becomes more precise. In the case of tourism, there is still a mismatch of regional positioning, but markets and policies can guide the diffusion and return of tourism factors through DI. Usually factors of production are biased to flow to

regions with a higher and more developed level of tourism factor allocation. The flow of dominant factors such as tourism capital, resources, and enterprises, dominated by market and planning mechanisms, can be achieved through DI to achieve top-level matchmaking and matching of tourism factors and market players. This further extends the innovativeness of market subjects and the flexibility of tourism factor allocation, and achieves the optimization of tourism factor allocation and efficiency. For example, the intelligent tourism recommendation system can recommend tourism-related information to market players in a targeted manner [48].

**3.3.2 Promote macro regulation and control.** Data can flow across time and space without wear and tear in the economic society, possessing a large amount of information and providing an important basis for rational decision-making by individuals, enterprises, and the state. Government departments are using data as an important basis to achieve public management and macro-control. Tourism destination governments use the available data information, learn from the experience of other regions or countries, establish tourism macroeconomic policies based on big data, take the initiative to promote the transformation of policies into tourism practices, enhance the scientific and rational nature of tourism macro policies, and help promote the tourism industry to high-quality development. In the era of increasing data transparency, tourism approaches and public services are also becoming more and more diverse [20]. Faced with this new situation, the government sector makes use of DI to share accurate and timely information related to tourism industry strategies, tourism economic operations, tourism construction guidelines, and market entry rules to give full play to the macro guidance potential of government departments. For example, the Authority uses data technology to predict how climate changes affect tourism destinations and provides response guidelines for public services, tourism transportation, and resource protection through a big data monitoring platform to enhance external response capabilities. As an important contributor to the efficiency of China's economic growth [49], DI can also collect, integrate, and process credit information, build a credit system for the whole society, and interconnect with other core sectors to create a favorable market environment for the tourism industry and to ensure the effectiveness of market management. At the same time, access to authentic information can mitigate information asymmetries and pre-existing market failures.

In summary, the macro mechanism of DE facilitating HQDT is shown schematically in Fig 3.

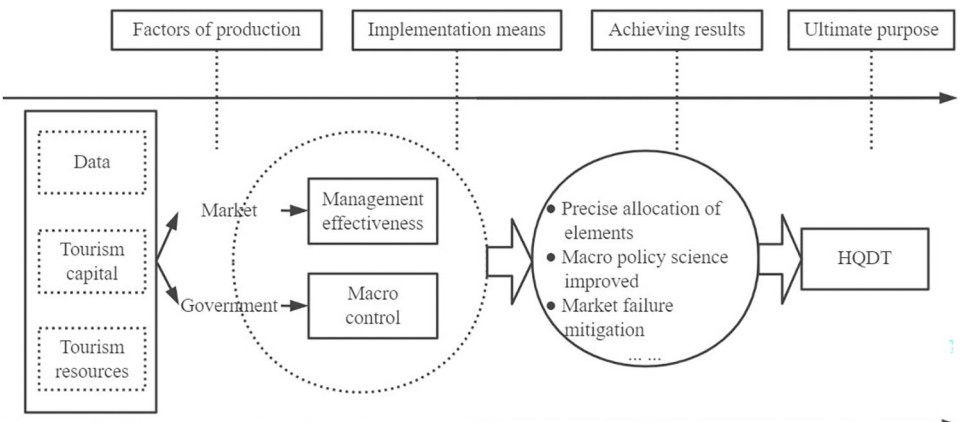

**Fig 3. A schematic diagram of the macro mechanism of the DE in promoting the HQDT.**

Based on the theoretical mechanisms explained in the above "micro-meso-macro" analytical framework, the following research hypotheses are proposed:

**Hypothesis 5.** The DE significantly and positively affects the HQDT.

# 4. Research design

The above theoretical analysis shows that the DE can contribute to HQDT at the micro, medium, and macro levels, respectively. In order to further explore the impact of the DE on tourism quality and to test the research hypotheses 1–5 proposed in the theoretical analysis, further empirical level tests are conducted in this paper.

## 4.1 Index system

With reference to Sun et al. [50], Sun et al. [51], Lu and Ren [52], and considering the availability of data and the hierarchy of measurement indicators, this paper constructs the evaluation index system of DE development level containing 20 measurement indicators in 4 subsystems of DI, digital industrialization, industrial digitization, and digital innovation capability (Table 1); and the evaluation index system of HQDT level containing 19 measurement indicators in 5 subsystems of innovation-driven, coordinated development, green development, internal and external opening and people's livelihood sharing (Table 2). Based on Sun et al. [51], Wang et al. [29], and Fang and Huang [53], this paper uses a coupled model to calculate the coordination between tourism and city, economy and environment. The tourism system

**Table 1. Evaluation index system of DE development level.**

| Target layer | Tier 1 indicators | Tier 2 indicators | Properties | Indicator meaning |
|---|---|---|---|---|
| DE System | DI | Internet broadband access ports | + | Broadband Internet base |
| | | Number of web pages | + | Digital application |
| | | Number of domains | + | Digital communications |
| | | Optical cable line length | + | Digital network foundation |
| | | Total length of cable broadcasting and television transmission trunk network | + | Digital media base |
| | Digital industrialization | Digital TV subscribers | + | Digital media applications |
| | | Cell phone penetration rate | + | Mobile Internet foundation |
| | | Information transmission, software and technology service industry employment | + | Information industry foundation |
| | | Software revenue | + | Industrialization of digital technology |
| | | Internet-related output | + | The Internet industry |
| | | Number of Internet-related employees | + | The Internet human resources elements |
| | | Total telecom business as a proportion of GDP | + | Telecom industry output |
| | | Internet broadband access users | + | Internet broadband applications |
| | Industrial digitization | Proportion of e-commerce purchases to GDP | + | E-commerce industry development |
| | | The proportion of the number of enterprises with e-commerce transaction activities | + | Enterprise digitalization |
| | | E-commerce sales to GDP share | + | E-commerce industry development |
| | | Digital inclusive finance index | + | Digital finance level |
| | Digital innovation ability | Number of patents granted | + | Innovation ability |
| | | R&D expenditure of industrial enterprises above scale | + | Investment in scientific research and innovation |
| | | Total technology market turnover | + | Digital innovation level |

**Table 2. HQDT level evaluation index system.**

| Target layer | Tier 1 indicators | Tier 2 indicators | Properties | Indicator meaning |
|---|---|---|---|---|
| HQDT level | Innovation-driven | Investment intensity of R & D funds | + | Innovation input |
| | | Number of domestic patent applications granted | + | Innovation capability |
| | | Number of tourism employees | + | Human capital |
| | Coordinated development | Growth rate of total tourism revenue | + | Internal coordination development |
| | | Share of tourism revenue in tertiary industry | + | Level of tourism coordination |
| | | Tourism industry and city coordination | + | Harmonized development of tourism and cities |
| | | Tourism industry and economic coordination | + | Harmonized development of tourism and economy |
| | | Tourism industry and ecological environment coordination degree | + | Harmonized development of tourism and ecology |
| | Green development | Investment completed in the treatment of waste gas and wastewater | + | Tourism pollution control level |
| | | Disposal rate of urban domestic waste | + | Domestic waste treatment capacity |
| | | Greening coverage rate of built-up areas | + | Greening coverage capacity of built-up areas |
| | | Number of nature reserves | + | Green development capacity of nature reserves |
| | Internal and external opening | The proportion of inbound tourism to the total number of tourists | + | External communication degree |
| | | Tourism foreign exchange revenue accounted for the proportion of total tourism revenue | + | Tourism foreign trade openness |
| | | A-class scenic spots | + | Tourism advanced development |
| | | Number of travel agencies | + | Cultural communication level |
| | Sharing the livelihood of the people | General public service expenditure | + | Shared service level |
| | | Passenger turnover by road, water transport and railroad | + | Tourism transportation Sharing level |
| | | Number of people employed in tourism/total employment | + | Tourism employment sharing level |

consists of the total tourism revenue, the number of tourism employees, the number of travel agencies, the number of star-rated restaurants, and the number of A-class scenic spots. The urban system selects three indicators of urban road area per capita, rental cars, and urban water penetration rate. The economic system selects three indicators of tertiary industry/GDP, GDP per capita, and the proportion of urban population. The environmental system consists of the greening coverage of built-up areas, the completed investment in the treatment of exhaust gas, and the completed investment in the treatment of wastewater.

## 4.2 Level measurement

The original data of the relevant indicators come from the 2011–2020 China Statistical Year-book, China Tourism Statistical Yearbook, China Environmental Statistical Yearbook, China Science and Technology Statistical Yearbook, the National Bureau of Statistics, and the statistical yearbooks of each province as well as the statistical bulletin of national economic and social development. The missing values in the indicators of 31 provinces for a few years were filled with the help of the average method. Drawing on Yao et al. [54], this paper uses the entropy method to measure the indicators.

The 2011–2020 scores of the 31 provinces for each dimension of the DE are shown in Table 3. In terms of the dimensions of the DE, the average scores (average of total scores) of the 31 provinces for DI, digital industrialization, industrial digitization, and digital innovation capability for 2011–2020 are 0.039, 0.055, 0.019, and 0.032, respectively. It can be seen that

**Table 3. Average scores for each dimension of the DE in 31 provinces, 2011–2020.**

| Region | DI | Region | Digital industrialization | Region | Industry digitization | Region | Digital innovation capability |
|---|---|---|---|---|---|---|---|
| National | 0.039 | National | 0.055 | National | 0.019 | National | 0.032 |
| Beijing | 0.116 | Beijing | 0.163 | Beijing | 0.057 | Beijing | 0.095 |
| Guangdong | 0.108 | Guangdong | 0.151 | Guangdong | 0.053 | Guangdong | 0.088 |
| Jiangsu | 0.086 | Jiangsu | 0.121 | Jiangsu | 0.042 | Jiangsu | 0.070 |
| Zhejiang | 0.070 | Zhejiang | 0.098 | Zhejiang | 0.034 | Zhejiang | 0.057 |
| Shandong | 0.069 | Shandong | 0.096 | Shandong | 0.034 | Shandong | 0.056 |
| Shanghai | 0.058 | Shanghai | 0.081 | Shanghai | 0.028 | Shanghai | 0.047 |
| Sichuan | 0.050 | Sichuan | 0.070 | Sichuan | 0.024 | Sichuan | 0.041 |
| Fujian | 0.046 | Fujian | 0.064 | Fujian | 0.022 | Fujian | 0.038 |
| Hubei | 0.039 | Hubei | 0.055 | Henan | 0.019 | Hubei | 0.032 |
| Henan | 0.038 | Henan | 0.053 | Hubei | 0.019 | Henan | 0.031 |
| Hebei | 0.035 | Hebei | 0.049 | Hebei | 0.017 | Hebei | 0.029 |
| Liaoning | 0.034 | Liaoning | 0.047 | Liaoning | 0.016 | Liaoning | 0.028 |
| Hunan | 0.033 | Hunan | 0.046 | Hunan | 0.016 | Hunan | 0.027 |
| Shaanxi | 0.031 | Shaanxi | 0.044 | Anhui | 0.015 | Shaanxi | 0.026 |
| Anhui | 0.030 | Anhui | 0.042 | Shaanxi | 0.015 | Anhui | 0.025 |
| Jiangxi | 0.029 | Jiangxi | 0.040 | Tianjin | 0.014 | Jiangxi | 0.024 |
| Tianjin | 0.028 | Tianjin | 0.039 | Jiangxi | 0.014 | Tianjin | 0.023 |
| Chongqing | 0.028 | Chongqing | 0.039 | Chongqing | 0.014 | Chongqing | 0.023 |
| Guizhou | 0.026 | Guizhou | 0.036 | Guizhou | 0.013 | Guizhou | 0.021 |
| Yunnan | 0.026 | Yunnan | 0.036 | Yunnan | 0.013 | Yunnan | 0.021 |
| Guangxi | 0.025 | Guangxi | 0.035 | Heilongjiang | 0.012 | Heilongjiang | 0.020 |
| Heilongjiang | 0.024 | Heilongjiang | 0.034 | Guangxi | 0.012 | Guangxi | 0.020 |
| Shanxi | 0.023 | Shanxi | 0.032 | Shanxi | 0.011 | Shanxi | 0.019 |
| Jilin | 0.022 | Inner Mongolia | 0.030 | Jilin | 0.011 | Jilin | 0.018 |
| Inner Mongolia | 0.021 | Jilin | 0.030 | Inner Mongolia | 0.010 | Inner Mongolia | 0.017 |
| Hainan | 0.021 | Hainan | 0.029 | Hainan | 0.010 | Hainan | 0.017 |
| Qinghai | 0.021 | Qinghai | 0.029 | Qinghai | 0.010 | Qinghai | 0.017 |
| Ningxia | 0.020 | Xinjiang | 0.028 | Ningxia | 0.010 | Ningxia | 0.016 |
| Xinjiang | 0.020 | Gansu | 0.027 | Xinjiang | 0.010 | Xinjiang | 0.016 |
| Gansu | 0.019 | Ningxia | 0.027 | Gansu | 0.009 | Gansu | 0.015 |
| Xizang | 0.017 | Tibet | 0.024 | Xizang | 0.008 | Xizang | 0.014 |

development of digital industrialization and DI is better than the development of industrial digitization and digital innovation capacity.

The overall scores of the 31 provinces on the level of DE and HQDT from 2011–2020 are shown in Table 4. Overall, the level of DE and HQDT is relatively higher in places with more fully developed economies, such as Beijing, Guangdong, Jiangsu, and Zhejiang. However, the overall level of 31 provinces is not high, and there are great differences between regions. Among them, the DE development level scores are all below 0.5, with a maximum difference of 0.4. The HQDT level scores are all below 0.5 except for Guangdong Province, with a maximum difference of 0.6. Specifically, the eastern region has a relatively high level of economic development, which can provide strong economic support, tap sufficient technical talents, ensure the improvement of the industrial base, popularize the operation of digital technology, and enhance the overall development environment, making the overall score of DE and tourism development more competitive. The "digital divide" between the Midwest and the East still exists objectively.

**Table 4. Results of measuring the level of DE and HQDT in 31 provinces.**

| Overall average score of DE (2011–2020) | | Overall average score of HQDT (2011–2020) | |
|---|---|---|---|
| Province | Score | Province | Score |
| Beijing | 0.431 | Guangdong | 0.670 |
| Guangdong | 0.399 | Jiangsu | 0.488 |
| Jiangsu | 0.320 | Shandong | 0.451 |
| Zhejiang | 0.260 | Zhejiang | 0.438 |
| Shandong | 0.255 | Beijing | 0.387 |
| Shanghai | 0.213 | Shanghai | 0.309 |
| Sichuan | 0.185 | Fujian | 0.306 |
| Fujian | 0.170 | Anhui | 0.290 |
| Hubei | 0.146 | Hunan | 0.289 |
| Henan | 0.140 | Sichuan | 0.289 |
| Hebei | 0.130 | Yunnan | 0.281 |
| Liaoning | 0.125 | Liaoning | 0.278 |
| Hunan | 0.121 | Hebei | 0.268 |
| Shaanxi | 0.116 | Henan | 0.265 |
| Anhui | 0.112 | Jiangxi | 0.253 |
| Jiangxi | 0.107 | Shaanxi | 0.253 |
| Chongqing | 0.104 | Hubei | 0.251 |
| Tianjin | 0.103 | Inner Mongolia | 0.243 |
| Yunnan | 0.096 | Guangxi | 0.239 |
| Guizhou | 0.095 | Shanxi | 0.231 |
| Guangxi | 0.092 | Guizhou | 0.214 |
| Heilongjiang | 0.090 | Heilongjiang | 0.203 |
| Shanxi | 0.086 | Tianjin | 0.197 |
| Jilin | 0.081 | Xinjiang | 0.195 |
| Inner Mongolia | 0.079 | Chongqing | 0.190 |
| Hainan | 0.078 | Hainan | 0.163 |
| Qinghai | 0.077 | Jilin | 0.159 |
| Ningxia | 0.073 | Gansu | 0.150 |
| Xinjiang | 0.073 | Xizang | 0.120 |
| Gansu | 0.070 | Qinghai | 0.082 |
| Xizang | 0.063 | Ningxia | 0.068 |

## 4.3 Variable description

As shown in Table 5, the explanatory variable of this study is the HQDT level, and the core explanatory variable is the level of DE development; the sub-dimensional explanatory variables under the core explanatory variables include DI, digital industrialization, industrial digitization, and digital innovation capacity. Considering that other factors can also have an impact on the HQDT in each province, this study adds relevant control variables to the benchmark model to improve accuracy. Drawing on the research of Lu and Ren [52], the control variables are as follows: (1) Total economic volume: The total economic volume has a crucial influence on the HQDT. (2) Economic growth rate: Economic growth is not only the basis for the HQDT, but also can effectively pull the HQDT. (3) Degree of openness: The level of openness to the outside world has a competitive and technological introduction effect, and therefore affects the HQDT. (4) Industrial structure change: Industrial structure change can have a regional universal and lasting impact on tourism development, and the rationalization of

**Table 5. Names and measures of each variable.**

| Variable type | Variable name | Variable symbols | Measurements indicate |
|---|---|---|---|
| Explained variables | Level of HQDT | Quality | HQDT composite score |
| Core explanatory variables | Level of DE development | Digiecon | DE development composite score |
| Sub-dimensional explanatory variables under the core explanatory variables | DI | Foundation | DI measurement score |
| | Digital industrialization | Industrialization | Digital industrialization measure score |
| | Industrial digitization | Digitalization | Industrial digitalization score |
| | Digital innovation capability | Innovation | Digital innovation capability measurement score |
| Control variables | Total economic volume | Lngdp | Logarithm of GDP by province |
| | Economic growth rate | $Gdp_{rate}$ | Growth rate of GDP by province |
| | Degree of external opening | Open | Ratio of total imports and exports to GDP by province |
| | Industrial structure change | Upgrade | Ratio of value added of tertiary industry to total value added of industry |
| | Scale of tourism industry | Scale | Ratio of total tourism industry revenue to GDP by province |
| | Government fixed asset investment | Invest | Investment in fixed assets-growth over the previous year |

industrial structure becomes a fundamental driving force for tourism development. (5) Industry scale: The greater the proportion of tourism industry, the more disposable expenditure the nation can invest in the tourism industry. (6) Fixed asset investment: The fixed asset investment policy introduced by the government is conducive to the HQDT. Table 6 gives the results of descriptive statistics for each variable. And there is no multicollinearity in each variable after passing the VIF test. Fig 4 shows the scatter plot and fitted curve of the DE and HQDT, which intuitively shows that there is preliminary evidence of a relatively obvious positive correlation between the DE and HQDT. However, the correlation needs to be further tested by more rigorous econometric models.

## 4.4 Model setting

**4.4.1 Benchmark regression model.** In this paper, in order to study the impact of DE on the HQDT, the benchmark regression model is developed with reference to Hao et al. [55] as follows:

$$Y_{it} = \alpha_0 + \alpha_1 X_{it} + \alpha Z + \omega_{it} \tag{1}$$

**Table 6. Results of descriptive statistics for each variable.**

| Variables | Sample size | Average | Median | Std | Maximum | Minimum |
|---|---|---|---|---|---|---|
| Quality | 310 | 0.265 | 0.250 | 0.130 | 0.780 | 0.049 |
| Digiecon | 310 | 0.145 | 0.110 | 0.113 | 0.629 | 0.006 |
| Foundation | 310 | 0.039 | 0.029 | 0.030 | 0.170 | 0.002 |
| Industrialization | 310 | 0.055 | 0.042 | 0.043 | 0.238 | 0.002 |
| Digitalization | 310 | 0.019 | 0.015 | 0.015 | 0.083 | 0.001 |
| Innovation | 310 | 0.032 | 0.024 | 0.025 | 0.139 | 0.001 |
| Lngdp | 310 | 9.695 | 9.818 | 1.000 | 11.619 | 6.416 |
| $Gdp_{rate}$ | 310 | 0.097 | 0.093 | 0.052 | 0.260 | -0.053 |
| Open | 310 | 0.297 | 0.153 | 0.321 | 1.586 | 0.008 |
| Upgrade | 310 | 0.494 | 0.486 | 0.089 | 0.837 | 0.327 |
| Scale | 310 | 0.178 | 0.152 | 0.094 | 0.735 | 0.039 |
| Invest | 310 | 10.469 | 10.050 | 11.032 | 40.600 | -56.600 |

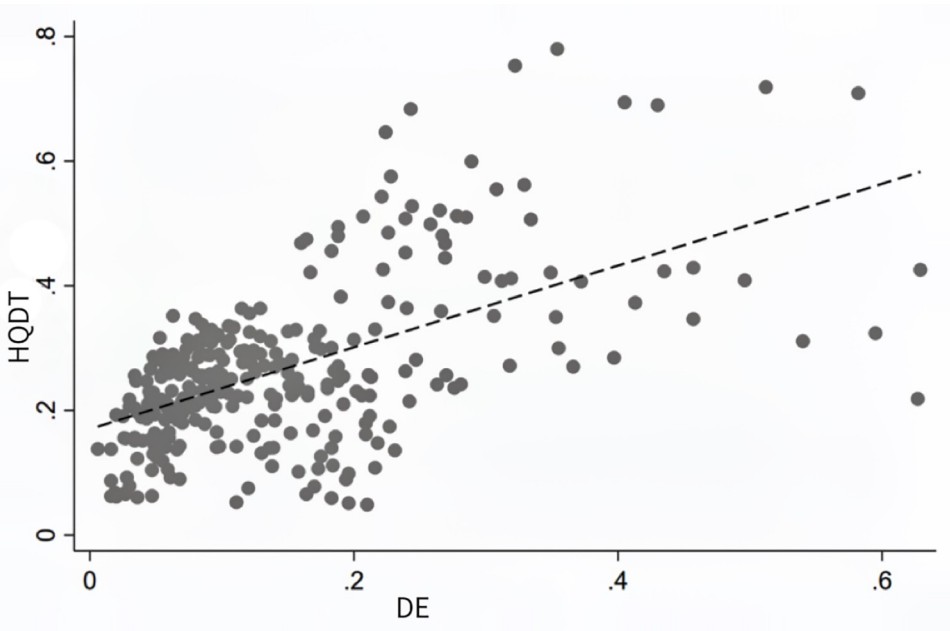

**Fig 4. Scatter plot and fitted curve of DE and HQDT.**

where $Y_{it}$ represents the explanatory variable; $X_{it}$ represents the core explanatory variable; $Z$ represents a series of control variables; $\alpha_0$ represents the constant term; $\omega_{it}$ represents the random error term; the subscript $i$ represents the province; $t$ represents the year.

**4.4.2 Spatial Durbin model (SDM).** SDM is able to observe the effect of lagged factors of the dependent variable on the explanatory variables, taking into account the role of spatial spillover effects of different factors on the explanatory variables. Referring to the research methodology used by Hao et al. [56] and Xue et al. [57], the SDM is:

$$Y_{it} = \rho W Y_{it} + \beta X_{it} + \theta W X_{it} + \gamma Z + \mu_i + v_t + \varepsilon_{it} \tag{2}$$

Where: $Y$ is the explanatory variable; $X$ is the explanatory variable; $W$ is the spatial weight matrix; $WY_{it}$ and $WX_{it}$ denote the spatial lag term; $\beta$ is the vector of parameters to be estimated for $X$; $Z$ stands for the ensemble of the control variables; $\mu$ and $v$ represent the individual and time fixed effects; $\varepsilon$ represents the randomized disturbance term.

Since the SDM encompasses effects on neighboring areas, the interpretation of its coefficients is not homogeneous. Therefore, further effect decompositions are performed in this paper [58]. The SDM of Eq (2) is recast in the following form:

$$Y_{it} = (1 - \rho W)^{-1}(\beta X_{it} + \theta W X_{it} + \gamma Z) + (1 - \rho W)^{-1}(\mu_i + v_t + \varepsilon_{it}) \tag{3}$$

## 5. The empirical results analysis

### 5.1 Benchmark regression results

Columns (2) (3) (4) (5) in Table 7 show the regression results for DI, digital industrialization, industrial digitization, and digital innovation capacity under fixed effects, all of which have a positive impact on HQDT, and hypotheses 1–4 have been verified. In the perspective of DE sub-dimension, perfect DI, solid digital industrialization, mature digitalization of industry, and super digital innovation capability can provide solid technical support, complete digital platform, good digital innovation atmosphere, and creative integration opportunities for the

**Table 7. Baseline regression results of the DE on HQDT.**

| Variables | (1) | (2) | (3) | (4) | (5) |
|---|---|---|---|---|---|
| Digiecon | 0.6744***(0.0623) | | | | |
| Foundation | | 2.4978***(0.2309) | | | |
| Industrialization | | | 1.7835***(0.1650) | | |
| Digitalization | | | | 5.1109***(0.4688) | |
| Innovation | | | | | 3.0569***(0.2827) |
| Constant term | -0.1425*(0.0741) | -0.1420*(0.0742) | -0.1429*(0.0741) | -0.1426*(0.0738) | -0.1420*(0.0742) |
| Control variables | Controlled | Controlled | Controlled | Controlled | Controlled |
| Region | Uncontrolled | Uncontrolled | Uncontrolled | Uncontrolled | Uncontrolled |
| Year | Controlled | Controlled | Controlled | Controlled | Controlled |
| $R^2$ | 0.8214 | 0.8213 | 0.8212 | 0.8221 | 0.8212 |
| Adj—$R^2$ | 0.8117 | 0.8115 | 0.8114 | 0.8124 | 0.8115 |
| F | 178.09 | 177.92 | 177.83 | 178.91 | 177.88 |

Note: Standard errors in parentheses

***, **, * represent statistical significance at the 1%, 5%, and 10% levels, respectively.

HQDT. The regression results of core variables under fixed effects are given in column (1) of Table 7, which shows that after controlling for other variables and fixed effects, the regression coefficient of the DE on HQDT is significantly positive, and hypothesis 5 is verified. The application of DE has not only reduced the management cost of tourism industry, but also accelerated its transformation and upgrading. The segmentation of tourists and precise matching realized by digital technology in tourism industry improves the efficiency of resource and factor allocation, optimizes tourism destination management and services, and satisfies tourists' personalized needs. By promoting cross-border integration and collaborative innovation in the tourism industry, the Internet can achieve effective development of tourism destination resources and green development, and effectively change the homogenous status quo of the tourism industry so that tourists can feel a better tourism experience. The tourism industry can guide the proliferation and return of tourism elements based on digital platforms, as well as use big data monitoring platforms as an important basis to enhance its external response capabilities. In short, the DE helps the tourism industry to achieve complementary economic and social benefits and promotes HQDT as a whole.

## 5.2 Spatial effects results

Considering the existence of spatial correlation between DE and HQDT level among different regions, this paper adopts spatial econometric model to investigate the possible spatial effects in the influence relationship. First, this paper uses the Moran index to test the spatial correlation [59]. The results of the spatial correlation test are shown in Table 8, which shows that all indices are significantly positive except for 2018, indicating that the DE has a positive spatial correlation with the level of HQDT. This indicates the need for further analysis of spatial effects.

Second, to test which spatial econometric model is applicable, this study conducted LM tests [60] (Table 9). The p-values of the spatial error effect (SEM) and spatial lag effect (SAR) tests are significant and positive, indicating that the selection of both the SEM model and the SAR model is appropriate. Therefore, the SDM that combines both is selected for this study. The Hausman test was also conducted in this study, which indicated that the time fixed effects

**Table 8. Spatial correlation test between DE and HQDT.**

| Year | Moran's I | Z-value | Year | Moran's I | Z-value |
|------|-----------|---------|------|-----------|---------|
| 2011 | 0.039** | 2.160 | 2016 | 0.058*** | 2.720 |
| 2012 | 0.047** | 2.380 | 2017 | 0.063*** | 2.898 |
| 2013 | 0.073*** | 3.118 | 2018 | 0.016 | 1.612 |
| 2014 | 0.035** | 2.069 | 2019 | 0.017* | 1.644 |
| 2015 | 0.043** | 2.315 | 2020 | 0.031* | 1.864 |

model should be used more appropriately. Therefore, the SDM time fixed effects model was chosen for the spatial effects test.

Finally, Referring to the bias correction procedure of Lee and Yu [61]: If the SAR, SEM, SDM, and SDEM models contain temporal fixed effects but not spatial fixed effects, bias correction can be performed with the help of parameter estimation (obtained by direct method). The regression results of the spatial effect of the DE on the HQDT are shown in Table 10. The coefficients after time fixed effects bias correction are still positive at the 1% significance level, indicating that there is indeed a significant spatial effect of the DE on the HQDT.

In order to further explore the specific spatial interaction effects of the DE on the HQDT, this study decomposes its spatial effects under the time fixed effects SDM [62, 63]. The coefficients of the direct effects in Table 11 are significantly positive, indicating that an increase in the level of the DE significantly and positively contributes to the HQDT in the region. The coefficient of the indirect effect in Table 11 is 0.1530, which is less significant but still under the 10% level. This indicates that the level of HQDT is also affected by the level of DE development in other provinces. The total effect in Table 11 passes the test at the 1% level of significance, indicating that one unit of change in the DE in all regions can have an impact of 0.8220 units on eco-efficiency in the region. In sum, the results in Table 11 show that the level of HQDT in China's provinces is affected not only by the level of development of the local DE, but also by the level of development of the DE in other spatially relevant provinces. In other words, the DE has significant spatial spillover effects on HQDT.

## 5.3 Endogeneity test and robustness test

**5.3.1 Endogeneity test.** Endogeneity problems may result if the higher the level of capital, technology, and innovation in a region when the quality of tourism development is high, i.e., if it is conducive to promoting the development of the DE in that region. In this regard, based on the availability of data, the number of fixed telephone year-end and mobile Internet users in each province were selected as instrumental variables [31] to address the endogeneity issue. The reasons for this are: First, the number of fixed telephone year-end and the number of mobile Internet subscribers are not directly related to the level of HQDT. Second, they are closely related to the DE. Third, they are not directly related to the control variables. The original hypothesis that the explanatory variables are all exogenous is rejected by adding the two instrumental variables to the Hausman test, proving that the original model is better with the addition of the two instrumental variables. Then the results are shown in the column of

**Table 9. Applicability tests of spatial measurement models.**

| Tests | Statistics | P | Tests | Statistics | P |
|-------|-----------|---|-------|-----------|---|
| LM spatial-error | 41.799 | 0.000 | LM spatial-lag | 9.592 | 0.000 |
| Robust LM spatial-error | 32.212 | 0.000 | Robust LM spatial-lag | 0.006 | 0.940 |

**Table 10. Regression results of the spatial effect of DE on the HQDT.**

| Variables | Time fixed effects | Time fixed effect bias correction |
|---|---|---|
| Digiecon | 0.6790***(0.0766) | 0.6790***(0.0741) |
| Lngdp | 0.0426***(0.0060) | 0.0426***(0.0058) |
| Gdp$_{rate}$ | -0.0423(0.1218) | -0.0423(0.1179) |
| Open | 0.1312***(0.0177) | 0.1312***(0.0171) |
| Upgrade | -0.4943***(0.0696) | -0.4943***(0.0674) |
| Scale | 0.2136***(0.0422) | 0.2136***(0.0408) |
| Invest | 0.0006(0.0004) | 0.0006(0.0004) |
| Region | Uncontrolled | Uncontrolled |
| Year | Controlled | Controlled |
| R$^2$ | 0.6609 | 0.6609 |

Table 12 (1) below, and the results show that the positive correlation between DE and HQDT still holds.

**5.3.2 Robustness test.** In the previous study, it was found that the DE has a positive impact on the HQDT, so this section uses three ways to conduct robustness tests, namely, shrinkage treatment, replacement of explanatory variables, and removal of some samples. Firstly, the continuous variables are shrunken in order to avoid estimation bias due to extreme values, and the results are shown in the columns of Table 12 (2). Secondly, column of Table 12 (3) adopts the results of DE development in each province measured by Du and Lou [64] to replace the comprehensive score of DE development level in this paper and re-estimate the model to further ensure the robustness of the empirical results in this paper. Finally, Table 12 (4) takes into account that the development of DE will be affected by the 2015 China stock market crash and the new crown pneumonia epidemic in 2020, and draws on the research practice of Wu et al. [65] to kick out the crisis factor for later testing. The robustness tests of all three approaches indicate that the DE has a positive contribution to the HQDT.

## 5.4 Heterogeneity analysis

Due to the differences in digital technology and human resources in different regions of China, differing from the East—Central—West region as the grouping standard that has been mostly used in the existing studies, this paper carries out regional grouping based on the scale of the provinces and the level of development of DE in order to further analyze the differential impact of DE of heterogeneous subjects on the HQDT.

**5.4.1 Heterogeneity analysis by province size.** Since there are differences in the scale of each province, the DE at different provincial scales may have heterogeneous impacts on the HQDT. In this paper, the provinces are divided into large-scale and small-scale provinces

**Table 11. Effect decomposition results of the SDM.**

| Variables | Direct effect | Indirect effect | Total effect |
|---|---|---|---|
| Digiecon | 0.669***(0.061) | 0.153*(0.090) | 0.822***(0.125) |
| Lngdp | 0.049***(0.005) | -0.045***(0.008) | 0.004(0.009) |
| Gdp$_{rate}$ | -0.036(0.131) | 0.156(0.0239) | 0.120(0.219) |
| Open | 0.119***(0.019) | 0.122***(0.032) | 0.240***(0.032) |
| Upgrade | -0.421***(0.071) | -0.623***(0.098) | -1.044***(0.122) |
| Scale | 0.254***(0.042) | 0.049(0.075) | 0.303***(0.077) |
| Invest | 0.0003(0.0004) | -0.0001(0.001) | 0.0002(0.001) |

**Table 12. Results of endogeneity test and robustness test.**

| Variables | Instrumental variables | Tailoring | Substitution of explanatory variables | Removal of partial samples |
|---|---|---|---|---|
| Digiecon | 1.3169***(0.1729) | 0.6966***(0.0619) | 0.0004***(0.0001) | 0.7679***(0.0685) |
| Lngdp | 0.0057(0.0097) | 0.0452***(0.0056) | 0.1009***(0.0051) | 0.0431***(0.0060) |
| Gdp$_{rate}$ | -0.1324(0.1332) | 0.0255(0.1239) | 0.0833(0.1383) | -0.0718(0.1374) |
| Open | 0.0875**(0.0404) | 0.1090***(0.0173) | 0.1278***(0.0194) | 0.0961***(0.0180) |
| Upgrade | -0.7480***(0.1480) | -0.4168***(0.0669) | -0.0886(0.0692) | -0.3823***(0.0725) |
| Scale | 0.3615***(0.0623) | 0.2260***(0.0459) | 0.0147(0.0448) | 0.2112***(0.0455) |
| Invest | 0.0001(0.0004) | 0.0006(0.0005) | 0.0008(0.0005) | 0.0004(0.0005) |
| Constant term | 0.3925***(0.1354) | -0.1488**(0.0717) | -0.7315***(0.0678) | -0.1261(0.0778) |
| Region | Uncontrolled | Uncontrolled | Uncontrolled | Uncontrolled |
| Year | Controlled | Controlled | Controlled | Controlled |
| R$^2$ | 0.7565 | 0.8301 | 0.7836 | 0.8375 |

based on the median year-end resident population of the province. From the regression results in columns (2) (3) in Table 13, it is concluded that the DE has a significant positive impact on the HQDT, regardless of whether it is a small-scale or large-scale province.

**5.4.2 Heterogeneity analysis by level of DE.** Considering that different provinces have different levels of DE, this paper divides the sample into high and low levels of DE according to the average of the comprehensive score of DE, and explores the heterogeneous influence of different levels of DE on the HQDT [36]. The results of the test are shown in (4) (5) in Table 13 below, where regions with high levels of DE significantly and positively affect the HQDT, while the effect of low-level regions is not significant. The reason for the existence of this may be that there is a great correlation between the development of the DE and the macro-economy, and that the low-level regions have more severe financial constraints as well as less efficient services compared to the regions with high-level DE, resulting in the lack of tourism resource development, tourism investment level, and product innovation. At the same time, tourism information dissemination in low-level areas is constrained, which makes it more expensive to reach consumers with tourism services and products. Coupled with the lower disposable income in low-level areas, which greatly inhibits tourism consumption demand, the impact of the DE on the level of HQDT is limited.

**Table 13. Results of heterogeneity analyses.**

| Variables | Provincial scale | | DE level | |
|---|---|---|---|---|
| | Small scale | Large scale | Low level | High level |
| Digiecon | 0.2059**(2.3600) | 0.3420***(3.4400) | 0.0577(0.1620) | 0.5246***(0.1079) |
| Lngdp | 0.0531***(7.7300) | 0.1053***(6.4700) | 0.0634***(0.0050) | 0.1747***(0.0287) |
| Gdp$_{rate}$ | 0.1336(1.1400) | -0.0536(-0.2700) | 0.1826*(0.0973) | -0.4237(0.3543) |
| Open | 0.0409*(1.8600) | 0.1554***(6.8500) | 0.0071(0.0193) | 0.1485***(0.0286) |
| Upgrade | 0.0572(0.5500) | 0.8086***(5.8800) | -0.0080(0.0805) | -0.3589**(0.1378) |
| Scale | 0.1441***(3.3100) | 0.3213***(4.7200) | 0.1444***(0.0337) | 0.3813*(0.2160) |
| Invest | -0.0001(-0.2300) | 0.0013**(2.0400) | 0.0001(0.0003) | 0.0019(0.0015) |
| Constant term | -0.3875***(-4.1600) | -1.3004***(-7.2400) | -0.4301***(0.0677) | -1.5230***(0.3465) |
| Region | Uncontrolled | Uncontrolled | Uncontrolled | Uncontrolled |
| Year | Controlled | Controlled | Controlled | Controlled |
| Observations | 160 | 150 | 220 | 90 |
| R$^2$ | 0.7668 | 0.9023 | 0.7308 | 0.8945 |

## 6. Discussion

This study examined the impact of the DE and its sub-dimensions on the HQDT, and the relevant conclusions were generally consistent with the proposed hypotheses. The study found that the DE has a significant positive impact on HQDT, which is in line with the findings of Yan [1]. Yan found that the DE contributes to the HQDT as a deepening and evolution of the three structures of "technology system-economic industry-social system", and concluded that the technology economy paradigm is the theoretical basis for the enabling effect of both [1]. Based on this, this study builds a clear theoretical framework from micro to meso to macro levels, extending the theoretical foundation of the DE for HQDT. Further, this study finds that DI supports the development of DE, which is in line with Zhang et al. [66]. They argue that DI investment plays a crucial role in strengthening the development framework of the DE, and based on this, this paper extends the analysis of the important role of DI for the HQDT. At the same time, this paper finds that industrial digitization in the DE drives the HQDT. Dang also discusses the transformation and upgrading of tourism and shows that the underlying logic of tourism digital transformation is to transform tourism productivity from being driven by traditional production factors to being driven by data factors and digital platforms, and that tourism can use digital transformation to solve the industry's existing quality and efficiency dilemmas. In contrast, this study focuses more on the impact of the tourism industry on its quality development with the help of digital transformation. This study also found that digital industrialization can contribute to HQDT, and Chen [67] and Yang [8] also conducted an exploration of digital industrialization. They indicated that digital industrialization, as a core industrial part of the DE, can achieve efficiency in the operation of the digital economic system. Based on this, this paper develops a study on the enhancement of digital industrialization on the high-quality operation of tourism. In addition, this study finds that the DE can promote tourism industry integration through digital innovation, which is consistent with the findings of Wu et al. [68] and Shen [69]. They found that DE and technological innovation can promote the integration of cultural tourism industry. This topic was also discussed by Mäkitie et al. [70], who emphasized the empowering role of digital innovation for sustainability transformation. This paper, for its part, emphasizes the role of digital innovation as a driver for HQDT. This study continues to explore the spatial effects of the DE on the HQDT and finds that the DE not only benefits the high-quality development of the local tourism industry but also acts on other spatially relevant provinces through spatial spillover effects. This is consistent with the findings of Wu et al. [71], who found a significant positive spatial association between the DE and the HQDT. Finally, this study found that the significant positive effect of DE on HQDT is not influenced by the size of the province but by the level of DE development. The heterogeneity has also been discussed by many scholars [72] and found that the DE significantly acts on the HQDT in the East-Central-West region, except for the Northeast region. In contrast, this study does not follow the traditional East-Central-West grouping of existing studies, but focuses on the provincial scale and the level of DE development, which extends the study of HQDT heterogeneity to some extent.

## 7. Conclusions and policy recommendations

### 7.1 Conclusions

Through theoretical analysis and empirical tests, this paper shows that the DE can contribute to HQDT in all aspects. ① At the micro level, the DE can help improve the efficiency of tourism enterprises, and its economy of scale and the Matthew effect can realize the reduction of average cost; its economy of scope can meet the diversified needs of consumers; and its long-

tail effect can perfect the matching mechanism of supply and demand. ② At the meso level, DE can realize the transformation and upgrading of the tourism industry structure through industry digitization and digital industrialization, and form a new tourism industry form and value chain through cross-border integration. ③ At the macro level, the DE can promote the innovation and flexibility of market players, increase the input of new factors in tourism, as well as improve their allocation efficiency, and further contribute to the macro regulation and management of the tourism market.  At the empirical level, the test results show that the DE significantly and positively affects the HQDT, and its DI, digital industrialization, industrial digitization, and digital innovation capacity have significant positive effects on the HQDT. ⑤ The significant positive effect of DE on HQDT is not affected by the size of the province but by the level of DE development, i.e., there is no significant promotion of HQDT in low level DE areas. ⑥ The DE shows a significant spatial spillover effect on the HQDT.

## 7.2 Policy recommendations

Following the discussion in this paper, the DE has helped the tourism industry move toward high-quality development mainly through the following paths:

① Promote the construction of DI. The flourishing development of DE cannot be achieved without a complete DI. Based on the current foundation, we should step up the coverage of existing infrastructure such as 5G base stations and fiber-optic cable lines, continuously improve the degree of construction of new infrastructure such as fiber-optic broadband and sensing terminals, as well as continuously promote the development process of convergent infrastructure such as vehicle networking and industrial Internet, to provide a favorable digital foundation for the HQDT. At the same time, the DI should be used to open up the information arteries of society, achieve precise positioning, and effectively improve the matching between the supply and dynamic demand for tourism.

② Accelerate the transformation of tourism industry digitalization. In the period of rapid development of DE, digital transformation is an important opportunity for tourism industry to transform and upgrade. However, unhealthy situations such as blind pursuit of digitalization, superficial digital transformation, and over-eager digital transformation should be avoided. Policy pilots for digital transformation in tourism should be strengthened, and a proper evaluation index system should be established to summarize the achievements and shortcomings of the pilots and address the barriers to transformation in a targeted manner. The tourism sector should also be stimulated to use popular industries such as web animation and Internet video to combine traditional tourism resources with modern popular elements to achieve innovative transformation of tourism.

③ Strengthen the development model of integration and innovation supported by digital technology. With the help of digital technology, the tourism industry can realize in-depth and innovative integration with various industries, which is an important way to promote HQDT in the era of DE. In macro strategy formulation, we should give full play to the guiding role of the market and the management function of the government. In micro strategy selection, we should find the fitting point of integration according to the change in tourism market demand, and build a diversified integration model through diversified integration paths. The creative integration of tourism with agriculture, sports, culture, and other industries should be facilitated. It should also actively strengthen the integration of the "DE + tourism", helping to connect Internet enterprises and tourism enterprises, to achieve digital innovation of traditional attractions, hotels and other tourism spaces, and to improve the level of intelligent services in tourism.

④ Overcome the serious challenges faced by tourism enterprises. On the one hand, because tourism enterprises are intermediary service enterprises, financing constraints have become a common problem for small and medium-sized tourism enterprises. Financial institutions should be encouraged to establish a lending system based on big data profiling and artificial intelligence, which not only reduces the cost of loan approvals for tourism enterprises but also increases the speed of lending, further preventing small and medium-sized tourism enterprises from facing bankruptcy as a result of slow liquidity in the short term. Tourism enterprises should also be encouraged to join the "Micro Service Cloud" and "Trusted Infrastructure" to establish a digital inclusive financial ecosystem for tourism enterprises, which can alleviate the financing difficulties of tourism enterprises. On the other hand, existing small and medium-sized tourism enterprises are unable to adapt to the pace of transformation and development in the digital economy due to their low level of technological application, resulting in their transformation being stalled. Small and medium-sized tourism enterprises should participate in the integration of industry and education in higher vocational colleges and universities. Through the deep integration of production and education with higher vocational colleges and research institutes, digital talents and digital technology can be introduced to enhance the level of digital technology application within the enterprise. The management structure of small and medium-sized tourism enterprises should also be changed for digital transformation, make full use of digital technology to optimize all kinds of enterprise resource allocation, and streamline the operation process, to promote the digital transformation of enterprises.

## 7.3 Limitations and further research

Although this study selected 19 sub-indicators to measure the level of HQDT in each province, the definition of HQDT is broad, so it is difficult to comprehensively cover its influencing factors during the study. Thus, how to accurately measure the level of HQDT is still a work that needs to be studied in depth in the future. In addition, this study only takes China as an example. Therefore, future research can consider the differences in the HQDT in different countries to conduct more studies. Finally, since the HQDT is a long-term project, it is a future research direction to explore the impact of DE on the long-term sustainability of HQDT. And, as digital technology advances, studying the role of emerging technologies such as artificial intelligence and blockchain in the tourism industry could be a valuable direction for future research.

## Supporting information

**S1 Data.**
(XLS)

## Acknowledgments

The authors would like to thank the reviewers and editors, as well as others who helped with the manuscript and whose suggestions greatly improved the manuscript.

## Author Contributions

**Conceptualization:** Min Yu.

**Data curation:** Min Yu, Binbin Ma.

**Formal analysis:** Min Yu, Dan Liu.

**Funding acquisition:** Binbin Ma.

**Investigation:** Min Yu.

**Methodology:** Min Yu, Aixia Zhang.

**Project administration:** Binbin Ma.

**Software:** Min Yu, Dan Liu.

**Supervision:** Binbin Ma.

**Validation:** Min Yu, Aixia Zhang.

**Writing – original draft:** Min Yu.

**Writing – review & editing:** Min Yu, Binbin Ma, Dan Liu, Aixia Zhang.

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
