## [Decision Letter · Decision Letter 0]

20 Feb 2024

PONE-D-24-04314Is the digital economy empowering high-quality tourism development? A theoretical and empirical research from ChinaPLOS ONE

Dear Dr. yu,

Thank you for submitting your manuscript to PLOS ONE. After careful consideration, we feel that it has merit but does not fully meet PLOS ONE’s publication criteria as it currently stands. Therefore, we invite you to submit a revised version of the manuscript that addresses the points raised during the review process.

We look forward to receiving your revised manuscript.

Kind regards,

Saliha Karadayi-Usta, PhD

Academic Editor

PLOS ONE

Journal Requirements:

3. PLOS requires an ORCID iD for the corresponding author in Editorial Manager on papers submitted after December 6th, 2016. Please ensure that you have an ORCID iD and that it is validated in Editorial Manager. To do this, go to ‘Update my Information’ (in the upper left-hand corner of the main menu), and click on the Fetch/Validate link next to the ORCID field. This will take you to the ORCID site and allow you to create a new iD or authenticate a pre-existing iD in Editorial Manager. Please see the following video for instructions on linking an ORCID iD to your Editorial Manager account: https://www.youtube.com/watch?v=_xcclfuvtxQ.

Reviewers' comments:

Reviewer's Responses to Questions

**Comments to the Author**

1. Is the manuscript technically sound, and do the data support the conclusions?

Reviewer #1: Partly

Reviewer #2: Yes

Reviewer #3: Yes

2. Has the statistical analysis been performed appropriately and rigorously? 

Reviewer #1: Yes

Reviewer #2: Yes

Reviewer #3: Yes

3. Have the authors made all data underlying the findings in their manuscript fully available?

Reviewer #1: Yes

Reviewer #2: Yes

Reviewer #3: Yes

4. Is the manuscript presented in an intelligible fashion and written in standard English?

Reviewer #1: Yes

Reviewer #2: Yes

Reviewer #3: Yes

5. Review Comments to the Author

Reviewer #1: The article "Is the Digital Economy Empowering High-Quality Tourism Development? A Theoretical and Empirical Research from China" presents a comprehensive examination of how the digital economy (DE) influences the high-quality development of tourism (HQDT) in China. The study explores the theoretical underpinnings of DE's role in HQDT and conducts empirical analysis using panel data from 31 provinces in mainland China over a decade. It highlights the positive impacts of DE on HQDT at micro, meso, and macro levels and suggests effective strategies for leveraging DE to promote HQDT.

Positive Aspects:

Theoretical Clarity: The article effectively clarifies the theoretical connotation of DE's enabling role in HQDT. By delineating DE's impact on efficiency improvements, industry structure transformation, and market innovation at micro, meso, and macro levels, the study provides a solid theoretical foundation for understanding the dynamics of DE in the tourism sector.

Comprehensive Empirical Analysis: The empirical analysis conducted using panel data from 31 provinces in China over a substantial time frame adds credibility to the study's findings. By examining the specific dimensions of DE's influence on HQDT and considering spatial spillover effects, the study offers valuable insights into the heterogeneous impact of DE on HQDT across different regions.

Policy Recommendations: The article concludes with actionable policy recommendations aimed at promoting DE's role in HQDT. Suggestions such as promoting digital infrastructure construction, accelerating tourism digital transformation, and alleviating financing constraints for tourism enterprises provide practical guidance for policymakers and industry stakeholders.

Areas for Improvement:

Lack of Comparative Analysis: While the study provides a thorough examination of DE's impact on HQDT in China, it would benefit from comparative analysis with other regions or countries experiencing similar digital transformations in the tourism sector. Comparative insights could offer a broader perspective on the effectiveness of DE in promoting HQDT beyond the Chinese context.

Limited Discussion on Challenges: Although the article outlines effective paths for leveraging DE to promote HQDT, it could delve deeper into the potential challenges and barriers faced by tourism enterprises in adopting digital technologies. Addressing these challenges and proposing strategies for overcoming them would enhance the practical relevance of the study's recommendations.

Future Research Directions: The article could expand on potential avenues for future research to advance understanding in this area. Identifying emerging trends, exploring the long-term sustainability of DE-driven HQDT initiatives, and investigating the role of emerging technologies such as artificial intelligence and blockchain in the tourism sector could be valuable directions for future research.

Overall, "Is the Digital Economy Empowering High-Quality Tourism Development? A Theoretical and Empirical Research from China" offers a valuable contribution to the literature on DE's impact on HQDT. By combining theoretical insights with empirical evidence and practical policy recommendations, the study enriches our understanding of the transformative potential of DE in the tourism sector while also highlighting avenues for further research and improvement.

Recommendations:

1. Data obtained from the China Bureau of Statistics and statistical yearbooks of various

provinces in China. How reliable are they?

2. 59 citations are not enough. Please add more.

E.g.

https://www.nature.com/articles/s41599-023-02150-7

and

https://gtg.webhost.uoradea.ro/PDF/GTG-3-2021/gtg.37309-711.pdf

Etc.

Reviewer #2: It suffers from several formal inconsistencies that need to be fixed. Apart from some minor suggestions on contents, particularly the formal structure of the paper needs thorough revision. Please also check typing matters (missing or superfluous blank spaces and blank lines, correct use of capital letters, get rid of similarity score (plagiarism), and so on).

Reviewer #3: The suggestions to the authors are included in a separate file. The file is attached. What the authors need to do is making some minor corrections to the presented manuscript.

And I suggest to check the References: perhaps a space is necessary after the publishin year.

6. PLOS authors have the option to publish the peer review history of their article (what does this mean?). If published, this will include your full peer review and any attached files.

Reviewer #1: No

Reviewer #2: **Yes: **Musallam R. Al-Rawahneh

Reviewer #3: **Yes: **Dr. Kamo Chilingaryan

---

## [Author Response · Author response to Decision Letter 0]

11 Mar 2024

Dear Editor, 

Thanks for your letter and reviewers' comments concerning our manuscript entitled "Is the digital economy empowering high-quality tourism development? A theoretical and empirical research from China". We thank you and the reviewers for the time and effort that you have put into reviewing the previous version of the manuscript. Those comments are all valuable and helpful for revising and improving our paper. We have studied all comments carefully and have made conscientious corrections. Revised portions are marked in the paper. We would like also to thank you for allowing us to resubmit a revised copy of the manuscript. The responses to the comments are as follows:

Reviewer 1

The article " Is the Digital Economy Empowering High-Quality Tourism Development? A Theoretical and Empirical Research from China" presents a comprehensive examination of how the digital economy (DE) influences the high-quality development of tourism (HQDT) in China. The study explores the theoretical underpinnings of DE's role in HQDT and conducts empirical analysis using panel data from 31 provinces in mainland China over a decade. It highlights the positive impacts of DE on HQDT at micro, meso, and macro levels and suggests effective strategies for leveraging DE to promote HQDT.

Positive Aspects: 

Theoretical Clarity: The article effectively clarifies the theoretical connotation of DE's enabling role in HQDT. By delineating DE's impact on efficiency improvements, industry structure transformation, and market innovation at micro, meso, and macro levels, the study provides a solid theoretical foundation for understanding the dynamics of DE in the tourism sector.

Comprehensive Empirical Analysis: The empirical analysis conducted using panel data from 31 provinces in China over a substantial time frame adds credibility to the study's findings. By examining the specific dimensions of DE's influence on HQDT and considering spatial spillover effects, the study offers valuable insights into the heterogeneous impact of DE on HQDT across different regions.

Policy Recommendations: The article concludes with actionable policy recommendations aimed at promoting DE's role in HQDT. Suggestions such as promoting digital infrastructure construction, accelerating tourism digital transformation, and alleviating financing constraints for tourism enterprises provide practical guidance for policymakers and industry stakeholders.

Areas for Improvement:

Comment 1: Lack of Comparative Analysis: While the study provides a thorough examination of DE's impact on HQDT in China, it would benefit from comparative analysis with other regions or countries experiencing similar digital transformations in the tourism sector. Comparative insights could offer a broader perspective on the effectiveness of DE in promoting HQDT beyond the Chinese context.

Response: We sincerely appreciate the valuable comment. Due to the limitations of the current data, we add this comment to "7.3 Limitations and Future Research: This study only takes China as an example. Therefore, future research can consider the differences in the high-quality development of tourism in different countries to conduct more studies." And in the next paper, we will delve deeper into the comparative analysis. 

Comment 2: Limited Discussion on Challenges: Although the article outlines effective paths for leveraging DE to promote HQDT, it could delve deeper into the potential challenges and barriers faced by tourism enterprises in adopting digital technologies. Addressing these challenges and proposing strategies for overcoming them would enhance the practical relevance of the study's recommendations.

Response: We really appreciate your comment on our manuscript. We have further examined the potential challenges and barriers faced by tourism businesses and proposed strategies to overcome them. Specifically in "7.2 Management Inspiration": ④Overcome the serious challenges faced by tourism enterprises. On the one hand, because tourism enterprises are intermediary service enterprises, financing constraints have become a common problem for small and medium-sized tourism enterprises. Financial institutions should be encouraged to establish a lending system based on big data profiling and artificial intelligence, which not only reduces the cost of loan approvals for tourism enterprises but also increases the speed of lending, further preventing small and medium-sized tourism enterprises from facing bankruptcy as a result of slow liquidity in the short term. Tourism enterprises should also be encouraged to join the "Micro Service Cloud" and "Trusted Infrastructure" to establish a digital inclusive financial ecosystem for tourism enterprises, which can alleviate the financing difficulties of tourism enterprises. On the other hand, existing small and medium-sized tourism enterprises are unable to adapt to the pace of transformation and development in the digital economy due to their low level of technological application, resulting in their transformation being stalled. Small and medium-sized tourism enterprises should participate in the integration of industry and education in higher vocational colleges and universities. Through the deep integration of production and education with higher vocational colleges and research institutes, digital talents and digital technology can be introduced to enhance the level of digital technology application within the enterprise. The management structure of small and medium-sized tourism enterprises should also be changed for digital transformation, make full use of digital technology to optimize all kinds of enterprise resource allocation, and streamline the operation process, so as to promote the digital transformation of enterprises.

Comment 3: Future Research Directions: The article could expand on potential avenues for future research to advance understanding in this area. Identifying emerging trends, exploring the long-term sustainability of DE-driven HQDT initiatives, and investigating the role of emerging technologies such as artificial intelligence and blockchain in the tourism sector could be valuable directions for future research.

Response: Thanks for your recommendation. Based on your suggestions, future research directions have been added under "7.3 Limitations and further research" by us: Finally, since high-quality development of tourism is a long-term project, it is a future research direction to explore the impact of digital economy on the long-term sustainability of high-quality development of tourism. And, as digital technology advances, studying the role of emerging technologies such as artificial intelligence and blockchain in the tourism industry could be a valuable direction for future research.

Overall, " Is the Digital Economy Empowering High-Quality Tourism Development? A Theoretical and Empirical Research from China" offers a valuable contribution to the literature on DE's impact on HQDT. By combining theoretical insights with empirical evidence and practical policy recommendations, the study enriches our understanding of the transformative potential of DE in the tourism sector while also highlighting avenues for further research and improvement.

Recommendations:

Comment 4: 1. Data obtained from the China Bureau of Statistics and statistical yearbooks of various

provinces in China. How reliable are they?

Response: Thanks for your recommendation. Data from the China Bureau of Statistics and China's various statistical yearbooks are official Chinese information and are considered by most to be very reliable. This type of data is widely used in academic research.

Comment 5: 2. 59 citations are not enough. Please add more. E.g. https://www.nature.com/articles/s41599-023-02150-7. And https://gtg.webhost.uoradea.ro/PDF/GTG-3-2021/gtg.37309-711.pdf Etc.

Response: We sincerely appreciate the valuable recommendation. We have scrutinized the literature and added more references related to the paper in the revised manuscript. Theses literature is as follows:

51. Yao Jiadai, Xu Pengpeng, Huang Zhijin. Impact of urbanization on ecological efficiency in China: An empirical analysis based on provincial panel data[J]. Ecological Indicators, 2021, 129, https://doi.org/10.1016/J.ECOLIND.2021.107827. 

52. Hao Xiaoli, Li Yuhong, Ren Siyu, et al. The role of digitalization on green economic growth: Does industrial structure optimization and green innovation matter?[J]. Journal of Environmental Management, 2023, 325(PA): 116504-116504, https://doi.org/10.1016/j.jenvman.2022.116504. 

53. Hao Yu, Liu Yiming, Weng Jia-Hsi, et al. Does the Environmental Kuznets Curve for coal consumption in China exist? New evidence from spatial econometric analysis[J]. Energy, 2016, 114, 1214-1223, https://doi.org/10.1016/j.energy.2016.08.075. 

54. Xue Dan, Yue Li, Ahmad Fayyaz, et al. Urban eco-efficiency and its influencing factors in Western China: Fresh evidence from Chinese cities based on the US-SBM[J]. Ecological Indicators, 2021, 127, https://doi.org/10.1016/J.ECOLIND.2021.107784. 

55. Ge Xiangyu, Zhou Zunrong, Zhu Xia, et al. The impacts of digital economy on balanced and sufficient development in China: A regression and spatial panel data approach[J]. Axioms, 2023, 12(2): 113-113, https://doi.org/10.3390/axioms12020113. 

56. Shen Xiaomeng, Zhao Haoxiang, Yu Jingyue, et al. Digital economy and ecological performance: Evidence from a spatial panel data in China[J]. Frontiers in Environmental Science, 2022, 10, https://doi.org/10.3389/FENVS.2022.969878. 

57. Yang Yangyang, Chen Weike, Gu Runde. How does digital infrastructure affect industrial eco-efficiency? Considering the threshold effect of regional collaborative innovation[J]. Journal of Cleaner Production, 2023, 427, https://doi.org/10.1016/J.JCLEPRO.2023.139248.

59. Zhang Zepu, Sun Chen, Wang Jing. How can the digital economy promote the integration of rural industries —— Taking China as an example[J]. Agriculture, 2023, 13(10), https://doi.org/10.3390/agriculture13102023. 

60. Chen Hanting, Ma Zhuoya, Xiao Hui, et al. The impact of digital economy empowerment on green total factor productivity in forestry[J]. Forests, 2023, 14(9), https://doi.org/10.3390/f14091729. 

61. Du Peng, Lou Feng. Research on the impact of digital economy development on the optimization and upgrading of industrial structure[J]. Business and Economic Research, 2022(18): 185-188.

Reviewer 2

Comment 1: It suffers from several formal inconsistencies that need to be fixed. 

Response: Thank you for your corrections. The form of the paper has been revised to be consistent in our resubmitted manuscript.

Comment 2: Apart from some minor suggestions on contents, particularly the formal structure of the paper needs thorough revision. 

Response: We really appreciate your comment on our manuscript. We have revised the core formal structure of the paper. In addition, during the restructuring process, we added more detailed spatial effects tests in "5.1 Analysis of Empirical Test Results" to make the spatial test results more rigorous and reliable. The study conclusions remain unchanged. The added methods are specified in "4.4.2 Spatial Durbin Model (SDM)" of the revised manuscript.

Comment 3: Please also check typing matters (missing or superfluous blank spaces and blank lines, correct use of capital letters, get rid of similarity score (plagiarism), and so on).

Response: Thank you for your careful scrutiny. In response to your comments, we have corrected missing or extra spaces and blank lines, correctly adjusted upper and lower case letters, as well as eliminated some similarity scoring.

Reviewer 3

Dear Authors

It is a great work. 

Comment 1: However, I suggest you including the names of sources more often than you did. This way we could know the scholars without the need to turn up/down pages.

Response: Thanks for your suggestion. Based on your suggestions, we have added more author names for the sources in the revised version.

The, some errors to correct:

Comment 2: Line 71- 73. Finally, the heterogeneity effect is fully considered, and the research sample is divided into different province sizes and different DE levels to be tested empirically respectively. In order that it can provide useful reference for the HQDT. (These two sentences should be joined)

Response: We really appreciate your comment on our manuscript. These two sentences have been merged as follows: Finally, this paper divides the research sample into different provincial sizes and different digital economy levels to fully test the effect of heterogeneity.

Comment 3: Line 168. multiple businesses or products.By effectively using the accumulated … (insert a space between the sentences)

Response: Thanks for your careful checks. The space has been added by us between sentences in the revised version.

Comment 4: Line 266. It uses his ITS strong drive and sense to produce correlation effect.

Response: Thank you for your correction. In our resubmission, "his" has been changed to "its". 

Comment 5: Line 295. through a big data monitoring platform to enhance external response capabilities. DI as a significant contributor to the efficiency of China's economic growth [46]. (Is it a sentence or a subheading?)

Response: Thank you for your careful scrutiny. It is the sentence, which has been modified and merged with the following sentence, as follows: As an important contributor to the efficiency of China's economic growth [46], DI can also collect, integrate, and process credit information, build a credit system for the whole society, and interconnect with other core sectors to create a favorable market environment for the tourism industry and to ensure the effectiveness of market management.

Comment 6: line 361. what is this word: industrial digitizationy.

Response: Thanks for your careful checks. Based on your comments, we have changed "digitizationy" to "digitization" in the revised version.

Comment 7: line 568. Although this study selected 19 sub-divisional indicators to measure the level of HQDT in each province. However,… (join these sentences)

Response: We sincerely appreciate the valuable comments. We have modified these two sentences as follows: Although this study selected 19 sub-indicators to measure the level of high-quality development of tourism in each province, the definition of high-quality development of tourism is broad, so it is difficult to comprehensively cover its influencing factors during the study.

In addition, We sincerely thank you for other valuable comments. We have revised the manuscript to comply with PLOS ONE style requirements. We have also provided data, verified ORCID iD, checked references, and uploaded images to PACE as required.

Thanks again to you and the reviewers for the time and effort on this paper. These comments are excellent and help us a lot to improve our articles! We sincerely hope that the revised manuscript will be accepted for publication in PLOS ONE.

---

## [Decision Letter · Decision Letter 1]

26 Mar 2024

PONE-D-24-04314R1Is the digital economy empowering high-quality tourism development? A theoretical and empirical research from ChinaPLOS ONE

Dear Dr. Ma,

Thank you for submitting your manuscript to PLOS ONE. After careful consideration, we feel that it has merit but does not fully meet PLOS ONE’s publication criteria as it currently stands. Therefore, we invite you to submit a revised version of the manuscript that addresses the points raised during the review process.

We look forward to receiving your revised manuscript.

Kind regards,

Saliha Karadayi-Usta, PhD

Academic Editor

PLOS ONE

Journal Requirements:

Reviewers' comments:

Reviewer's Responses to Questions

**Comments to the Author**

1. If the authors have adequately addressed your comments raised in a previous round of review and you feel that this manuscript is now acceptable for publication, you may indicate that here to bypass the “Comments to the Author” section, enter your conflict of interest statement in the “Confidential to Editor” section, and submit your "Accept" recommendation.

Reviewer #1: All comments have been addressed

Reviewer #3: (No Response)

2. Is the manuscript technically sound, and do the data support the conclusions?

Reviewer #1: Yes

Reviewer #3: Yes

3. Has the statistical analysis been performed appropriately and rigorously? 

Reviewer #1: Yes

Reviewer #3: Yes

4. Have the authors made all data underlying the findings in their manuscript fully available?

Reviewer #1: Yes

Reviewer #3: Yes

5. Is the manuscript presented in an intelligible fashion and written in standard English?

Reviewer #1: Yes

Reviewer #3: Yes

6. Review Comments to the Author

Reviewer #1: The authors revised and corrected their article. Almost perfect. I recommend some articles to cite. More than 70 citations are needed.

https://journals.sagepub.com/doi/10.1177/14673584231198414

and

https://www.emerald.com/insight/content/doi/10.1108/TRC-11-2022-0028/full/pdf

and

https://gtg.webhost.uoradea.ro/PDF/GTG-3-2021/gtg.37309-711.pdf

Etc.

Reviewer #3: Dear authors

I am afraid, there was some misunderstanding.

I asked the authors not to have digits (at least not as many) when referring to a source. E.g., If tourism enterprises are considered as the supply side, the AI in the growth

phase is influencing their employment, costs, and management practices [21].

Do the respected authors think that we are going to turn pages up and down to see that it was Wang yueying (by the way, why is the surname/name in lower case?)

Instead of this frustrating writing (NOT only here but in the entire text) the manuscript should have said: As Wang yueying states/mentions/claims/says/notices/emphasizes [21], if tourism enterprises are considered as the supply side, the AI in the growth phase is influencing their employment, costs, and management practices.

Lines 89 to 110 demands 13 times of up/down of pages. I will be surprised to see a person doing that, especially the author who you are citing in the manuscript without paying enough respect to his/her achievements.

7. PLOS authors have the option to publish the peer review history of their article (what does this mean?). If published, this will include your full peer review and any attached files.

Reviewer #1: No

Reviewer #3: **Yes: **Dr. Kamo Chilingaryan

---

## [Author Response · Author response to Decision Letter 1]

28 Mar 2024

Dear Editor and Reviewers,

Thanks for your letter concerning our manuscript entitled "Is the digital economy empowering high-quality tourism development? A theoretical and empirical research from China". We thank you for the time and effort that you have put into reviewing the previous version of the manuscript. Those comments are all valuable and helpful for revising and improving our paper. We have studied all comments carefully and have made conscientious corrections. Revised portions are marked in the paper. We would like also to thank you for allowing us to resubmit a revised copy of the manuscript. The responses to the comments are as follows:

Reviewer 1

The authors revised and corrected their article. Almost perfect.

Comment : I recommend some articles to cite. More than 70 citations are needed.

https://journals.sagepub.com/doi/10.1177/14673584231198414 and

https://www.emerald.com/insight/content/doi/10.1108/TRC-11-2022-0028/full/pdf and

https://gtg.webhost.uoradea.ro/PDF/GTG-3-2021/gtg.37309-711.pdf

Etc.

Response: Sincerely, thanks for your suggestion. Based on your suggestions, we have added your recommended articles in the revised version, as follows:

39. Roziqin Ali, Kurniawan Alferdo Satya, Hijri Yana Syafriyana, et al. Research trends of digital tourism: A bibliometric analysis[J]. Tourism Critiques: Practice and Theory, 2023, 4(1-2): 28-47, https://doi.org/10.1108/TRC-11-2022-0028.

45. Ives Gutierriz, João J Ferreira, Paula O Fernandes. Digital transformation and the new combinations in tourism: A systematic literature review[J]. Tourism and Hospitality Research, 2023(0), https://doi.org/10.1177/14673584231198414.

47. Setiawan Priatmoko, Lóránt Dénes Dávid. Winning tourism digitalization opportunity in the indonesia CBT business[J]. GeoJournal of Tourism and Geosites, 2021, 37(3): 800-806, https://doi.org/10.30892/gtg.37309-711.

Reviewer 3

Dear authors

I am afraid, there was some misunderstanding.

Comment : I asked the authors not to have digits (at least not as many) when referring to a source. E.g., If tourism enterprises are considered as the supply side, the AI in the growth phase is influencing their employment, costs, and management practices [21]. Do the respected authors think that we are going to turn pages up and down to see that it was Wang yueying (by the way, why is the surname/name in lower case?). Instead of this frustrating writing (NOT only here but in the entire text) the manuscript should have said: As Wang yueying states/mentions/claims/says/notices/emphasizes [21], if tourism enterprises are considered as the supply side, the AI in the growth phase is influencing their employment, costs, and management practices. Lines 89 to 110 demands 13 times of up/down of pages. I will be surprised to see a person doing that, especially the author who you are citing in the manuscript without paying enough respect to his/her achievements.

Response: We sincerely appreciate the valuable comments. We apologize for our misunderstanding. Based on your detailed comments and to fulfill the formatting requirements of PLOS ONE, we borrowed the citation format of Li and Yang (Li Hao, Yang Zihan. Does digital economy development affect urban environment quality: Evidence from 285 cities in China[J]. Plos One, 2024, 19(2): e0297503, https://doi.org/10.1371/journal.pone.0297503). When we cited the authors' achievements, we added the author's source to each citation. It is as follows: 

“At the theoretical level: Microscopically, digital intelligence technology can establish and maintain a good relationship between consumers and tourism enterprises by providing high quality services to further improve consumer satisfaction (Wang, 2022) [20]. If tourism enterprises are considered as the supply side, the AI in the growth phase is influencing their employment, costs, and management practices (Rodolfo and Chris, 2010) [21]. Dimitra et al. argue that big data can improve the efficiency, productivity, and profitability of tourism enterprises and can provide differentiated, rich, and convenient experiences to consumers (Dimitra et al., 2020) [22]. ” Etc.

In addition, we sincerely thank you for the valuable comments. We have also provided data, verified ORCID iD, checked references, and uploaded images to PACE as required by PLOS ONE.

Thanks again to the editor and the reviewers for the time and effort on this paper. These comments are excellent and help us a lot to improve our articles! We sincerely hope that the revised manuscript will be accepted for publication in PLOS ONE.

---

## [Decision Letter · Decision Letter 2]

8 Apr 2024

Is the digital economy empowering high-quality tourism development? A theoretical and empirical research from China

PONE-D-24-04314R2

Dear Dr. Ma,

We’re pleased to inform you that your manuscript has been judged scientifically suitable for publication and will be formally accepted for publication once it meets all outstanding technical requirements.

Kind regards,

Saliha Karadayi-Usta, PhD

Academic Editor

PLOS ONE

Additional Editor Comments (optional):

Reviewers' comments:

Reviewer's Responses to Questions

**Comments to the Author**

1. If the authors have adequately addressed your comments raised in a previous round of review and you feel that this manuscript is now acceptable for publication, you may indicate that here to bypass the “Comments to the Author” section, enter your conflict of interest statement in the “Confidential to Editor” section, and submit your "Accept" recommendation.

Reviewer #1: All comments have been addressed

Reviewer #3: All comments have been addressed

2. Is the manuscript technically sound, and do the data support the conclusions?

Reviewer #1: Yes

Reviewer #3: Yes

3. Has the statistical analysis been performed appropriately and rigorously? 

Reviewer #1: Yes

Reviewer #3: Yes

4. Have the authors made all data underlying the findings in their manuscript fully available?

Reviewer #1: Yes

Reviewer #3: Yes

5. Is the manuscript presented in an intelligible fashion and written in standard English?

Reviewer #1: Yes

Reviewer #3: Yes

6. Review Comments to the Author

Reviewer #1: The authors revised and corrected their article. I recommend to publish this in current form in Plos One journal.

Reviewer #3: Dear authors

As kind advice, next time you write a manuscript and you have someone to quote, use these expressions (not at once, please): as Li [2] considers/states/ mentions/emphasizes... or In his book Capital, K .Marx [36] admits that...., or Deng Xiaoping [17] indicates/mentions/argues that ....

7. PLOS authors have the option to publish the peer review history of their article (what does this mean?). If published, this will include your full peer review and any attached files.

Reviewer #1: No

Reviewer #3: **Yes: **Kamo Chilingaryan (PhD)

---

## [Editor Report · Acceptance letter]

22 Apr 2024

PONE-D-24-04314R2 

PLOS ONE

Dear Dr. Ma, 

I'm pleased to inform you that your manuscript has been deemed suitable for publication in PLOS ONE. Congratulations! Your manuscript is now being handed over to our production team.

Kind regards, 

on behalf of

Assoc. Prof. Dr. Saliha Karadayi-Usta 

Academic Editor

PLOS ONE